# Pathophysiology and Molecular Imaging of Diabetic Foot Infections

**DOI:** 10.3390/ijms222111552

**Published:** 2021-10-26

**Authors:** Katie Rubitschung, Amber Sherwood, Andrew P. Crisologo, Kavita Bhavan, Robert W. Haley, Dane K. Wukich, Laila Castellino, Helena Hwang, Javier La Fontaine, Avneesh Chhabra, Lawrence Lavery, Orhan K. Öz

**Affiliations:** 1Department of Radiology, University of Texas Southwestern Medical Center, 5323 Harry Hines Blvd, Dallas, TX 75390-8542, USA; katie.rubitschung@utsouthwestern.edu (K.R.); Amber.Sherwood@utsouthwestern.edu (A.S.); Avneesh.Chhabra@utsouthwestern.edu (A.C.); 2Department of Plastic Surgery, University of Cincinnati, 231 Albert Sabin Way, Cincinnati, OH 45267-0558, USA; crisolpa@ucmail.uc.edu; 3Department of Internal Medicine, Division of Infectious Diseases and Geographic Medicine, University of Texas Southwestern Medical Center, 5323 Harry Hines Blvd, Dallas, TX 75390-8542, USA; kavita.bhavan@utsouthwestern.edu (K.B.); laila.castellino@utsouthwestern.edu (L.C.); 4Department of Internal Medicine, Epidemiology Division, University of Texas Southwestern Medical Center, 5323 Harry Hines Blvd, Dallas, TX 75390-8542, USA; robert.haley@utsouthwestern.edu; 5Department of Orthopedic Surgery, University of Texas Southwestern Medical Center, 5323 Harry Hines Blvd, Dallas, TX 75390-8542, USA; dane.wukich@utsouthwestern.edu; 6Department of Pathology, University of Texas Southwestern Medical Center, 5323 Harry Hines Blvd, Dallas, TX 75390-8542, USA; helena.hwang@utsouthwestern.edu; 7Department of Plastic Surgery, University of Texas Southwestern Medical Center, 5323 Harry Hines Blvd, Dallas, TX 75390-8542, USA; Javier.lafontaine@utsouthwestern.edu (J.L.F.); Larry.lavery@utsouthwestern.edu (L.L.)

**Keywords:** diabetic foot infection, molecular imaging, test predictive value, X-ray, optical tomography, DWI, SPECT

## Abstract

Diabetic foot infection is the leading cause of non-traumatic lower limb amputations worldwide. In addition, diabetes mellitus and sequela of the disease are increasing in prevalence. In 2017, 9.4% of Americans were diagnosed with diabetes mellitus (DM). The growing pervasiveness and financial implications of diabetic foot infection (DFI) indicate an acute need for improved clinical assessment and treatment. Complex pathophysiology and suboptimal specificity of current non-invasive imaging modalities have made diagnosis and treatment response challenging. Current anatomical and molecular clinical imaging strategies have mainly targeted the host’s immune responses rather than the unique metabolism of the invading microorganism. Advances in imaging have the potential to reduce the impact of these problems and improve the assessment of DFI, particularly in distinguishing infection of soft tissue alone from osteomyelitis (OM). This review presents a summary of the known pathophysiology of DFI, the molecular basis of current and emerging diagnostic imaging techniques, and the mechanistic links of these imaging techniques to the pathophysiology of diabetic foot infections.

## 1. Introduction

In the past 30 years, diabetic foot infections (DFI) have become increasingly prevalent due to the rising incidence of diabetes mellitus (DM). According to the most recent National Diabetes Statistics Report by the Centers for Disease Control and Prevention, 30 million U.S. adults were reported to have diabetes, and 84.1 million were considered prediabetic [1]. Twenty percent of diabetes-related hospital admissions in the U.S. are from DFI [2], which is typically introduced by direct inoculation through a traumatic entry site in an insensate foot. Cellulitis, myositis, tendinitis, or OM occur in over half of all cases of diabetic foot ulceration (DFU) [3]. The climbing rate of diabetes is accompanied by an increased incidence of DFU, deep tissue infections, and amputations.

A global meta-analysis by Zhang et al. reported that DFU is prevalent in 6.3% of the population, with North America having the highest prevalence of DFU in the world (13.0%). This is followed by Africa (7.2%), Asia (5.5%), Europe (5.1%), and Oceania (3%), although a lack of studies led South America to be excluded from the meta-analysis. Countries with the highest prevalence of DFU included Belgium (16.6%), Canada (14.8%), and the United States (13.0%) [4]. The authors of this meta-analysis chose to exclude any studies that stated ulcer recurrence, suggesting that the overall occurrence of DFU is higher. Furthermore, these numbers are expected to continue rising due to widespread obesity and an aging population.

According to the same meta-analysis, the population most at risk for developing DFU were males with type 2 diabetes and low body mass index (mean BMI of 23.8 ± 1.7 in diabetic patients with DFU versus 24.4 ± 1.7 in diabetic patients without DFU), long diabetic duration (mean 11.3 ± 2.5 years in diabetic patients with DFU versus 7.4 ± 2.2 years in diabetic patients without DFU), diabetic retinopathy, and a history of smoking and hypertension [4]. A retrospective study by Wang et al. found a similar trend in Chinese patients: DFI was more common in men, and the disease prognosis worsened with age. On average, men experienced greater infection severity and underwent amputation 10 years earlier than women, who showed improved wound healing possibly attributable to higher estrogen levels [5,6].

Major amputations have also been associated with decreased life expectancy [7]. While this increased mortality is real, patients who undergo major amputation often die from complications of cardiovascular disease (myocardial infarction, heart failure, and cerebrovascular accidents). Prevention of major amputation is paramount because this can preserve ambulation and, consequently, cardiovascular fitness. Amputees often experience drastic lifestyle changes, including loss of independence and psychological distress. These factors are compounded in developing countries such as Ghana, where access to resources such as prosthetic services or mental health care is limited. [8]. Amputations are life-altering, as well as expensive. Studies have found that, in the United States, a lower limb amputation costs anywhere from $43,800 to $66,215 for the surgical procedure and aftercare such as rehabilitation services, nursing homes, and internal medicine costs [9]. Additionally, lower limb amputation places the contralateral limb at risk for complications due to increased functional demand.

The financial ramifications of DFI are also far-reaching. Collectively, $9 billion is spent annually on diabetic foot care in the U.S., which amounts to more than the yearly cost of breast or colorectal cancers [10]. Ulcers that culminate in DFI alone cost upwards of $1 billion every year in the U.S. [11]. The cost of medical treatment is tremendous, but does not account for indirect expenses related to loss of income, employment, or opportunity. Unfortunately, these secondary costs are often difficult to measure, and the true economic impact of DFI is likely to be much steeper than presently reported. Altogether, this information suggests that investing in the prevention of DFU and early detection of DFI would be economically advantageous.

Given the high prevalence, morbidity, and financial burden of diabetic foot infections, diagnostic methods that efficiently and accurately distinguish soft tissue infection from OM and monitor the therapeutic response are important for optimizing clinical outcomes. Distinguishing soft tissue infection from osteomyelitis may be challenging in clinical practice. Without these diagnostic features, diabetic patients are likely to experience exacerbated infection severity and poorer recovery prognosis. Herein, we review the current understanding of DFI pathogenesis and the corresponding mechanisms of applied imaging modalities.

## 2. Pathophysiology and Clinical Assessment of DFI

### 2.1. Diabetic Susceptibility to DFI

Several pathological factors place diabetic patients at increased risk for foot infections, including diabetic neuropathy, vascular insufficiency, and immunological dysfunction [12]. A strong link has been reported between diabetic neuropathy and foot ulceration [13]. Diabetic neuropathy is a complex polyneuropathy consisting of peripheral, motor, and autonomic neuronal damage [14].

Peripheral neuropathy is prevalent in 10.9–32.7% of diabetic patients in the U.S. [15,16]. Minor foot trauma from sources such as ill-fitting shoes or injury goes unnoticed from lack of sensation [17]. When repeated or left unattended, this trauma may lead to ulceration (Figure 1) and subsequent infection. Muscle atrophy, foot deformity (Charcot arthropathy, claw foot, pes cavus, hallux valgus), and gait abnormalities are caused by motor neuropathy resulting from the loss of myelinated fibers [18]. These biomechanical changes predispose neuropathic patients to foot ulceration due to increased pressure and shearing. Additionally, a study by Lung et al. suggests that a moderate-to-fast walking intensity decreases plantar stiffness and increases risk of foot ulceration compared to walking at slower speeds [19]. Charcot arthropathy develops in 13% of patients with neuropathy. It can result in fractures, dislocations, and fracture dislocations causing profound deformity and a rocker-bottom appearance of the plantar foot, due to bone breakdown and joint collapse [20,21]. Autonomic neuropathy in the lower limb causes vasodilation and excess warmth in the foot. Moreover, impaired neuronal control of the sweat glands reduces perspiration (anhidrosis), leading to dry skin that is prone to fissure or callus and increasing the risk of developing a foot wound [22,23]. Diabetic patients may also experience vascular insufficiency (micro and macrovascular) and immunological dysfunction, which further predisposes them to ulceration, impaired healing, and infection (Figure 1).

### 2.2. Clinical Diagnostic Tests

Several diagnostic tests are available to help with the clinical assessment of DFI, including the probe-to-bone test, biopsy, and assessment of inflammatory markers (ESR and CRP). The probe-to-bone test is routinely used during physical examination of the patient to evaluate the potential for OM [24]. In this procedure, a sterile tip probe is introduced through the ulcer to determine if bone is palpable. A solid/gritty end-point is considered positive for OM [25]. Suspected infections may be evaluated for microorganisms by sampling the wound site with an ulcer swab, soft tissue biopsy, or bone biopsy.

Many have argued that the results of these tests should not be independently used for diagnosis and should be only used as a screening method [26]. A negative test result in any of these cases does not exclude the diagnosis of DFI and a positive result may be misleading. For instance, soft tissue biopsies or swab cultures only assess the superficial flora and may not reflect the full extent of infection [27]. Bone biopsies are also limited because the technique requires trained personnel and it is prone to sampling errors. Moreover, there is no standardized definition of a positive bone biopsy and many classifications exist, each with its own merits and weaknesses—further convoluting the problem [27,28]. Historically, classification systems did not include the presence of OM as an indication of infection severity, and it is well known that the presence of OM negatively impacts the outcomes of diabetic foot infections [29]. Without a universal classification system, determinants of infection severity may vary, leading to a diagnostic disagreement among physicians. The accuracy of tests such as the probe-to-bone test relies on the prevalence of OM in the patient cohort being examined. For example, a group of hospitalized patients with severe infection may have a high positive predicative value using these methods, while a group of outpatients with low likelihood of OM may have a low positive predicative value [30]. These factors, as well as others, have encouraged advancements in understanding of the molecular mechanisms in DFI.

### 2.3. Molecular Mechanisms of DFI Features

All of the currently available diagnostic tests rely on the assessment of changes in DFI at a tissue level (Figure 2); however, such changes are only detectable when damage has already been inflicted. A strong understanding of the molecular changes that precede visual tissue damage is required to advance the diagnosis and treatment of DFI.

Bone deformities in the foot such as Charcot arthropathy occur when bone destruction from osteoclastic resorption exceeds bone building from osteoblastic recruitment. This breakdown, radiographically visualized as osteolysis, is mediated through the inflammatory RANKL (receptor activator of the nuclear factor-kappa B ligand) pathway and has been identified as an important pathway in the pathogenesis of Charcot foot. When RANKL concentration is higher than its competitive antagonist, OPG (osteoprotegerin), osteoclastogenesis is stimulated, affecting increased osteoclast activity, osteolysis, and deformity [31,32,33].

Arterial atherosclerosis, excessive leukocyte adhesion, increased vascular permeability, impaired hemostasis, and altered proliferation and apoptosis of vascular cells are all factors contributing to diabetic vascular insufficiency [34]. Poorly controlled glucose levels activate the polyol glucose metabolism pathway in which glucose is reduced to sorbitol and subsequently oxidized to fructose. Excess fructose and sorbitol lead to increased osmotic molality and ultimately osmotic stress [35]. Competitive consumption of NADPH and decreased nitric oxide synthetase activity also contribute to this challenge, as low nitric oxide concentration stimulates vessel constriction [36].

In addition to diminished perfusion, immune dysfunction has also been reported [37]. Diabetic patients frequently present with defective neutrophil function irrespective of glycemic status, including disrupted migration patterns associated with reduced production of chemotactic factors, amplified generation of reactive oxygen species, and abated phagocytosis from complement system dysfunction. Collectively, these changes facilitate pathogen invasion and infection development [38]. One retrospective clinical study demonstrated that the ratio of lymphocytes to neutrophils, platelets, and monocytes could predict the need for amputation due to DFI [39].

The innate immune system is a major source of cytokines. Macrophages and dendritic cells may produce cytokines in response to damage-associated molecular patterns (DAMPS) through the activation of pattern recognition receptors (PRRs). These inflammatory cytokines (such as TNFα, IL-1, IL-6, and interferon-ϒ) trigger osteoblast upregulation of RANKL, facilitating osteoclastogenesis [40]. Mitochondrial antiviral signal proteins (MAVS) also stimulate infected cells to secrete cytokines, activating the NF-κB and IRF3 pathways, which regulate the expression of type-I interferons. When bound to type-1 interferon receptors, these cytokines activate the JAK-STAT pathway, inhibiting pathogen replication and assembly due to a heightened expression of interferon-stimulated genes [41]. Type-I interferons such as INF-β are expressed in stromal cells, which are capable of differentiating into mature osteoblasts [42]. 

The combination of diabetic neuropathy, impaired perfusion, neutrophil dysfunction, and cytokine imbalance encourages infection development and progression, ischemic ulcers, or gangrene, which may culminate in amputation [22]. To avoid amputation and disease progression, a deeper understanding of pathogen involvement is necessary.

A wide variety of microorganisms may cause DFI (Figure 3), the most common of which is *Staphylococcus aureus*. The methicillin-resistant Staphylococcus aureus strain (MRSA) occurs in 16.78–30% of DFI cases, although this value varies geographically [43,44]. MRSA infection has been shown to have no impact on mortality, but is correlated with an increased rate of hospitalization and higher risk of limb amputation [45]. Amputation is effective in preventing the spread of infection, and studies in the United Kingdom and Germany have found that the procedure increased the life expectancy by 2 years in 50% of the diabetic subjects studied [46,47]. Even so, only 56% of diabetic patients with ulcerative infections were found to survive 5 years after initial onset of the ulcer [47]. Altogether, this information highlights the need for improved ulcer prevention and the prompt diagnosis of DFI. Imaging may define the precise location of the infected bone and establish the border for amputation. Advanced imaging methods may allow for early detection and verify the adequate response to therapy to avoid amputation.

## 3. Imaging Modalities

Imaging plays an important role in the diagnosis and treatment of DFI. Compared to the probe-to-bone test and biopsy, imaging offers non-invasive methods of detection that characterize many features associated with DFI. Advantages of imaging include the ability to specify infection sites, distinguish soft tissue infection from OM, identify bone abnormalities of Charcot arthropathy, evaluate neural or vascular disease, and characterize DFI pathology in extensive detail. A vital contribution of dual modality molecular and anatomic imaging is the ability to delineate the margin of infected bone from healthy bone, which can be used to determine the appropriate treatment course, including helping to define resection margins [48]. The American College of Radiology (ACR) has issued guidelines for musculoskeletal infections including DFI, soft tissue, or suspected DFO. The practice guidelines suggest screening with radiography, followed by Magnetic Resonance Imaging (MRI), and, finally, radionuclide-based cellular and molecular imaging methods as needed [49]. In this section, we discuss imaging modalities (Table 1) that are currently used to assess DFI, including radiography, MRI, computed tomography (CT), ultrasound, and angiography, with particular focus on molecular techniques.

### 3.1. Anatomical Modalities

#### 3.1.1. Radiography

Radiographs are the first-line modality for imaging DFI because of their low cost, high accessibility, and versatility. Radiography facilitates identification of bone changes associated with infection as well as changes in soft tissue (Figure 4).

Although conventional radiography is inexpensive, convenient, and fast, there is a delay in reaching the threshold of detection for visualizing OM. Clear evidence of bone erosion may not occur until 10–14 days after infection onset [50]. The radiographic soft tissue abnormality of DFI varies based on the type of infection and tissue response. Soft tissue infections involving gas gangrene or necrotizing fasciitis may be visible as radiolucent patchy gas bubbles in the fascial planes. Cellulitis is seen as the obliteration of fascial planes with stranding in the subcutaneous tissues, while abscess is seen as a focal dense lesion, however, the findings can be non-specific [77]. Visualization of radiolucent foreign bodies (e.g., wooden) can also be challenging on routine radiography [78].

Changes in bone architecture that occur in OM include regional osteopenia, periostitis, cortical bone loss, endosteal scalloping, structural changes in trabecular bone (demineralization), peripheral sclerosis, or peripheral new bone formation [22]. A study by Alvaro-Afonso et al. compared these imaging signs’ radiographic images to bone biopsy. Periosteal reaction, cortical disruption, affected bone marrow (defined as focal loss of trabecular pattern or marrow radiolucency), and sequestrum were evaluated. The most reliable and accurate feature was cortical disruption (sensitivity = 0.70–0.82, specificity = 0.38–0.57) and the least accurate and least agreed upon parameter among clinicians was periosteal reaction (sensitivity = 0.30–0.43, specificity = 0.67–0.84). When at least one of these four parameters was considered an indicator of OM, a pooled sensitivity of 0.86 and specificity of 0.27 was observed (PPV = 0.65–0.76, NPV = 0.36–0.62, PLR = 1.04–1.34, NLR = 0.33–0.80) [51]. Low specificity in radiography often leads to false positive diagnoses of OM, especially in patients with neuroarthropathic changes, such as Charcot’s foot [77]. A false positive diagnosis can result in prolonged antibiotic therapy and aggressive surgical resection of bone.

If a radiograph is strongly suggestive of OM, such as a foot ulcer with adjacent cortical erosion, further imaging may not be warranted. However, OM cannot be excluded on a negative radiograph given the limited sensitivity of radiography compared to other anatomic imaging modalities, such as MRI [79]. Despite this limitation, radiographs are helpful in detecting key features associated with DFI and can guide clinical decision making on further work-up.

#### 3.1.2. Ultrasonography 

Ultrasound (US) is a fast, inexpensive, safe, and highly accessible imaging technique that provides real-time images without the use of ionizing radiation [22]. After interacting with a tissue, the amplitude of reflected ultrasonic waves specific to a given tissue are computer analyzed and an image is produced. For DFI, frequencies very between 3–18 MHz; however, higher frequencies have limited tissue penetration and may not reach deeper tissues. Doppler US can be used to assess arterial blood flow velocity to peripheral extremities. Anechoic areas, such as fluid in abscesses, appear dark on US while hyperechoic tissues such as cortical bone are highly reflective and produce a bright signal. The range of gray scale helps identify bone, soft tissue, and fluid [79]. 

US is often used for draining collections of fluid and assessing arterial occlusions or lesions. Tenosynovitis and joint effusion can also be identified using US. The utility of US is limited by its low sensitivity (78%) and specificity (80%) for OM as well as shadowing caused by cortical bone [52,56]. Instead, the advancement of other imaging modalities such as CT, MRI, and Positron Emission Tomography (PET) has provided new opportunities in the assessment of DFO.

#### 3.1.3. Computed Tomography (CT)

CT imaging capitalizes on the concept that tissues of variable density differentially attenuate X-ray beams. This characteristic is captured in a parameter called the attenuation coefficient, expressed as the Hounsfield unit. CT is particularly useful in the assessment of bone and can detect osseous erosion sooner than radiography. Following image acquisition, computer algorithms generate multiplanar cross-sectional images. Bone images can be separated into the marrow compartment and the cortical compartment. Quantitative CT (qCT) provides compartment-specific bone mass and mineral density measurements. In a study by Commean et al., these qCT parameters were used to monitor changes in the midfoot bones of healthy control patients and type II diabetic patients with peripheral neuropathy. Volumetric qCT was able to detect osteolytic and osteopenic changes in the diabetic foot with comparable precision to that of dual X-ray absorptiometry (bone volume precision: 0.1–0.9%, BMD precision: 0.6–1.9%). Minor osteolytic or osteopenic changes were also detectable, rendering qCT a potentially useful tool in tracking the treatment response in DFI or neuropathic arthropathy with the development of appropriate biomarkers [80]. 

Clinically, CT provides rapid, high resolution image acquisition of a large anatomic region as well as quantifiable parameters, rendering it an excellent candidate for emergent diagnostic use [81]. Dual energy CT (DECT) enables the identification of bone marrow edema similar MRI. CT is often used to guide biopsy procedures. Some studies have reported that CT is a significant predictor of bone culture positivity [82], although there is a lack of consensus about this relationship in the literature [57,83]. DFI therapies such as wound debridement and abscess drainage can also be guided by CT. In neuropathic arthropathy, CT may be used for surgery planning and treatment monitoring [84]. 

Several features of OM are detectable by CT. These include soft tissue swelling from pro-inflammatory cytokine release by monocytes and other immune cells, cortical erosion from bacterial breakdown of bone, sinus tracts from the bone to the skin surface, decreased attenuation of the medullary space or bone marrow edema on DECT due to bone marrow inflammation, sequestrum of necrotic bone within the marrow, or gas accumulation from incomplete oxidation by anaerobic or facultative aerobic bacteria [85,86]. Contrast agents, which increase the mass attenuation coefficient of the tissue, can be coupled with CT imaging to enhance the visualization of soft tissue inflammation, and outline the wall of soft tissue or intra-osseous abscess. The small, iodinated molecules in the contrast dyes possess an X-ray absorbing capability, but also display nonspecific distribution and fast clearance; however, in patients with a contrast agent allergy, adverse events may occur, and a different technique may be needed.

In the diagnosis of DFI, CT is limited by a beam-hardening effect in the presence of implants or metal, which obstructs visibility and causes low soft tissue contrast. Moreover, CT uses ionizing radiation whereas MRI does not [79], and MRI possesses superior soft tissue contrast, sensitivity, and image resolution, making it a better candidate for the assessment of DFI [49,79,81]. 

#### 3.1.4. Magnetic Resonance

Following radiographic evaluation, MRI is used to further evaluate the presence, extent, and severity of DFI. MRI includes sequences of radiofrequency pulses and magnetic field gradients to evaluate tissue features. An MR exam may range in scanning duration, typically lasting 30–45 min, and spares the patient exposure to ionizing radiation. MRI is generally considered an anatomic method because it uses predominantly morphological sequences in clinical practice. Image contrast on morphological sequences depends on the density of proton spins at T1 and T2 relaxation times. T1 relaxation, also called the spin–lattice, is the time for longitudinal magnetization to return to the magnetic field axis (z) after radiofrequency pulses are applied. The spin–spin relaxation time (T2) refers to the time it takes for transverse magnetization to return to the transverse plane. By varying the contrast weighting toward T1 or T2, images can be generated with significant contrast between water and fat. For example, tissues with higher water content such as areas of edema appear dark on T1-weighted images and bright on T2-weighted images, compared to surrounding tissues. Tissues such as fat have the opposite appearance and appear bright on T1-weighted images and less bright on T2-weighted images [87]. However, morphological sequences exhibit limited characterization of soft tissue and bone pathology [88]. Molecular advancements such as dynamic contrast enhancement (DCE), diffusion weighted imaging (DWI), and Dixon-based fat suppression have improved the sensitivity and specificity of MR for the detection and characterization of neuropathic arthropathy and OM—unlike radiography, CT, or ultrasound [76,89].

Bone marrow edema is present in both neuropathic arthropathy and OM and is one pathologic feature that can be exploited by imaging applications. Increased white blood cell accumulation, inflammation, and higher marrow space permeability are characteristics associated with marrow edema in OM. Whereas, in neuropathic arthropathy, marrow edema tends to have lower cellularity and cell variety in comparison to OM [90]. Cortical erosion, ‘ghost sign’ (relative disappearance of bone on T1-weighted image as compared to T2-weighted image), ‘penumbra sign’ (high signal rim of abscess on T1-weighted image) are excellent MRI signs of OM versus cysts and well-defined margins of a neuropathic bone. In DCE-MR, a gadolinium-based contrast agent is injected after a baseline assessment, and multiple images are acquired over a few minutes to evaluate the microvascular environment (Figure 5i). The concentration of the contrast agent within tissues and blood vessels affects the signal intensity of MR. Ultimately, a time–concentration curve is generated which can be quantitatively analyzed to assess the physiological parameters of tissues and surrounding vasculature such as vessel permeability and perfusion [87,91,92]. In a prospective study by Liao et al., receiver–operating characteristic curves (ROCs) were used to assess the accuracy of several DCE-MR parameters in 30 diabetic foot patients with bone marrow edema caused by OM or neuropathic arthropathy. They found that patients with OM had higher contrast agent wash-in from the vascular to interstitial space (K^trans^), higher gadodiamide washout from the interstitial space back into the vascular space (K_ep_), and higher extra-axial extracellular space volume (V_e_) than patients with neuropathic arthropathy. ROC curves revealed that, of these parameters, K^trans^ and V_e_ were the best parameters for distinguishing OM from neuropathic arthropathy [90]. 

Another molecular technique used in the assessment of bone marrow edema is DWI, which uses the random (Brownian) movement of free water to generate an indirect estimation of cellularity and cell membrane integrity (Figure 5e,f). As the degree of diffusion weighting—also referred to as the b-value or diffusion moment—increases, the accuracy of DWI improves. The apparent diffusion coefficient (ADC) is the exponential decay of a single component of the diffusion signal which is often used to characterize lesions quantitatively (Figure 5g,h). In OM and soft tissue infection, inflammatory cells, pus, and cellular waste decrease free water diffusion, leading to high signal intensity and moderate-to-low ADC values in abscesses and nonviable tissue. In cases of pure edema—such as in neuropathic arthropathy—high signal intensity and high ADC values are seen due to the T2-shine through phenomenon [76,93]. A majority of the appendicular skeleton contains yellow marrow with mostly large adipocytes, resulting in reduced perfusion and lowered ADC values [94]. Since adipocyte content in the marrow space increases with age and can affect ADC values, fat suppression techniques should be used to further assess the bone marrow compartment [76]. 

One example of a fat suppression technique is chemical shift. Chemical shift separates fat and water signals according to their echo times (TEs) since they resonate at two different frequencies. The external magnetic field (B_0_) induces an electron current surrounding the nucleus, which leads to a locally produced magnetic field. [95]. When the locally produced field vector is opposite to that of the external field, it is termed opposed-phase; when the locally produced field vector is in the same direction as that of the external field, it is termed in-phase. The Dixon sequence combines the in-phase and out-of-phase images with decreased sensitivity to inhomogeneities of B_0_ and B_1_, resulting in more homogenous fat suppression and creation of multiple maps. Up to four contrasts may be acquired from one image, including in-phase (Figure 5c), opposed-phase (Figure 5d), water only (Figure 5b), and fat only. The Dixon sequence can also be used with other sequences such as gradient or fast spin echo and signal weighting [96]. Because of the homogenous fat suppression achieved by the Dixon technique, it is particularly useful in distinguishing bone marrow edema of OM from that of neuropathic arthropathy. The Dixon sequence is also used to aid in the identification of other features of OM including intraosseous sequestrums, fistulas, cortical destruction, and periostitis [97]. If both OM and neuropathic arthropathy are present, bone contours relatively disappear on T1-weighted images and reappear on fat-suppressed images (also known as the ghost sign). If only neuropathic arthropathy is present, true destruction of the bone occurs and the contours will not re-appear on fat-suppressed images [98,99]. A study by Fujii et al. assessed the efficacy of MR for DFO diagnosis and surgical margin selection. Using histopathology as the gold standard, and T1-weighted and fat-suppressed T2-weighted images, they found that MR was effective in diagnosing neuropathic ulcers (sensitivity and specificity 100%); however, it was not effective in diagnosing ischemic ulcers (sensitivity 29.6%). Bone marrow edema was not well visualized in cases of severe infection and ischemia, rendering MR ineffective in determining surgical margin selection [100]. 

Several MR techniques are used in the evaluation of peripheral nerves, which can be damaged by diabetes mellitus. Morphologic neurography includes T1-weighted, short-TI inversion recovery (STIR), and fat-suppressed T2-weighted sequences (Figure 5a). This series of images allows for morphological characterization of the nerves, soft tissue, and bone [101,102]. DWI neurographic sequences have also been used; however, DWI has limited spatial resolution, rendering the small nerves of the distal foot difficult to visualize. Diffusion tensor imaging (DTI) neurography is another technique used in the assessment of peripheral nerves and has been found to detect diabetic peripheral neuropathy in both type I and type II diabetics [103,104,105,106]. DTI uses the sensitivity of DWI to the anisotropic water movement within myelinated axons to generate high resolution images that can even provide information regarding myelin sheath damage caused by diabetes. Diabetic neuropathy treatment progression can be tracked using DTI neurography through changes in quantitative biomarkers such as fractional anisotropy, mean diffusivity, and radial diffusivity [76,107]. MR neurography can also aid in pre-surgical planning. One retrospective study used a combination of the techniques mentioned above, including T2-weighted images, T2-weighted non-fat-saturated images, and DWI, to determine the role of MR neurography in preoperative assessment. MR neurography was found to lead to a change in diagnosis or surgical decision in 21.4% of cases [108]. Even though the study was not exclusively performed on diabetics, it highlighted the range of cases that MR neurography can be applied to. 

Although these molecular applications of MR (Table 2) may be useful in the differentiation of OM from neuropathic arthropathy, nuclear imaging modalities such as ^18^F-FDG PET/CT have been reported to have higher diagnostic accuracy [109]. Moreover, MR cannot be used in patients with electronic or ferromagnetic implants, and patient movement can sometimes be problematic. Most importantly, although MR is an excellent anatomical method for imaging soft tissues, it does not provide direct information about molecular changes in the tissue.

### 3.2. Molecular Imaging

Radiotracers are chemical compounds containing a radioactive atom that may be injected alone or conjugated to a targeting moiety. The total concentration is so low that the underlying physiology is not disturbed by the radioimaging probe. After administration, molecular interactions with the radiotracer at sites of interest allow the metabolism of a tissue to be detected and quantified. Scintigraphy, single-photon emission computed tomography (SPECT), and PET imaging are molecular imaging modalities used for DFI assessment [110]. They possess higher sensitivity and specificity than anatomical modalities, translating to improved distinction between the infection of soft tissue and bone. Although molecular imaging modalities independently show promise in the evaluation of DFI, the combination of molecular imaging and anatomical studies may possess even greater sensitivity and specificity than either modality alone. For example, as mentioned above, the American College of Radiology listed MR as a first-line modality for cases of suspected OM because of its ability to produce high quality anatomical images [49]. However, the diagnostic accuracy of MRI may significantly improve when combined with molecular imaging assays such as PET, SPECT, or CT. Combined modality approaches provide direct information regarding molecular and tissue level activity. 

#### 3.2.1. Bone Scintigraphy

Planar bone scintigraphy involves intravenous injection of a radiotracer—typically ^99m^Tc-methylene diphosphonate or hydroxymethylene diphosphonate (^99m^Tc-MDP/HDP). Other radiotracers such as ^18^F-sodium fluoride (^18^F-NaF) may be used for PET imaging of bone. The diphosphonates are retained in bone by binding to hydroxyapatite crystals. Areas of greater radiotracer incorporation by osteoblasts indicate areas of high bone turnover and are often the focal point of infection. Triple phase bone scintigraphy involves capturing images at three stages: immediately after radiotracer injection, 10 min after radiotracer injection, and 2–4 h after radiotracer injection. Each image is used to assess different aspects of DFI. The first image provides information regarding blood flow to the region of interest. The second image, also called the tissue or blood pool phase, assesses the soft tissues and increased retention is generally associated with inflammation or infection. In the third phase, the amount of radiotracer uptake indicates the rate of new bone formation by osteoblasts (Figure 6). 

When assessing diabetic foot infection by three-phase bone scintigraphy, variation in radiotracer uptake is observed between phases. In cases of cellulitis and other soft tissue infections, increased radiotracer uptake is only seen during the first two phases, whereas, in OM, increased radiotracer uptake is seen in all three phases identifying focal hyperperfusion, hyperemia, and bone uptake. Moreover, because the third phase of bone scintigraphy reflects general bone turnover, bone deformities associated with degenerative joint disease, fracture, or orthopedic hardware can significantly reduce specificity [79,111]. A meta-analysis by Llewellyn et al. reported an average sensitivity of 84.2% with a relatively low specificity of 67.7% [52]. Although bone scintigraphy is highly sensitive, arterial occlusion or nonspecific inflammatory conditions such as Charcot joint may render the method susceptible to misdiagnosis in cases of DFI [22,52]. Impaired peripheral perfusion may lead to false negatives when assessing the diabetic foot. Sterile inflammation, fracture healing, or Charcot joint may cause increased radiotracer uptake during the third phase, which could be misidentified as a sign of increased bone turnover associated with OM, resulting in falsely positive interpretations. By utilizing a second scintigraphic modality, such as bone marrow imaging with a ^99m^Tc-labeled colloidal preparation and a tomographic method such as SPECT or SPECT/CT, specificity and anatomic localization are improved. This approach provides improved localization of areas with radiotracer uptake that may have been previously obstructed [79].

#### 3.2.2. Radiolabeled WBC and Bone Marrow Planar Scintigraphy, SPECT, and SPECT/CT

While planar bone scintigraphy uses radiotracers to directly detect areas of high bone turnover, leukocyte scans trace localization of radiolabeled white blood cells (WBC) that migrate to the site of infection. In this case, the WBCs act as carriers of the molecular imaging agent. The majority of the WBCs labeled by conventional methods are neutrophils, which are the first and most abundant leukocytes to arrive at the locus of infection [112]. As a result, WBC scans are better for acute infections and can generate highly specific images which can be further improved for spatial localization using SPECT or SPECT/CT. 

WBC radiolabeling is performed by in vitro or in vivo methods. In vitro WBC labeling involves the isolation of WBCs from fractionated blood, direct labeling with a radiotracer such as ^111^In-oxine or ^99m^Tc-HMPAO (exametazine), and reinjection into the subject. In vitro labeling can be limited by the concentration of recovered leukocytes, as optimal labeling requires at least 2000 leukocytes/μL [113]. ^111^In-oxine displays a normal distribution in the spleen, liver, and bone marrow and because it is highly stable, delayed imaging can be performed to allow labeled cells more time to localize to sites of infection (Figure 7). Importantly, there is no uptake in the uninfected foot. Most importantly, ^111^In-oxine-labeled cells can be combined with ^99m^Tc radiolabeled agents to target activated bone marrow in dual tracer studies. Such studies are useful in distinguishing OM from Charcot neuropathic changes (Figure 8). 

Unfortunately, ^111^In-oxine WBC scans produce low-resolution images and require a long interval between patient injection and image acquisition (16–30 h); however, ^99m^Tc-HMPAO (Figure 9) overcomes these problems by producing relatively high-quality images and requiring only a short interval between injection and image acquisition. Previous analyses have found that ^99m^Tc-HMPAO WBC scans are highly sensitive and specific in the detection of OM (91% and 92%, respectively) [58]. At present dual-tracer scans are not performed with ^99m^Tc WBC scans because no suitable marrow-targeting agents are commercially available for simultaneous imaging. 

WBC scans have also been used to assess the efficacy of antibiotic treatment in patients with DFO. Vouillarmet et al. [66] followed 45 patients with newly diagnosed DFO for either remission or treatment failure at one year. All patients were initially treated with appropriate antibiotics for six weeks, after which WBC-SPECT/CT was performed after infusion of ^99m^Tc-HMPAO-labeled leukocytes. In the 23 with negative scans, antibiotics were discontinued; whereas, in the 22 with positive scans, antibiotic therapy was continued for six more weeks after which a second scan was obtained. At the one-year clinical follow-up, 22 of the 23 with negative scans at 6 weeks remained in remission. The remaining 22 were re-scanned at 12 weeks after completion of the second course of antibiotics. At their one-year follow-up, all 9 with a negative re-scan remained in remission; whereas, of the 13 with positive scans at 12 weeks, 7 were in remission at 1 year, and 6 had failed treatment. Overall, in this two-stage testing and treating study, the sensitivity and specificity of the ^99m^Tc-HMPAO-labeled WBC-SPECT/CT were 82% and 86%, respectively, while the predictive value of a negative scan was 97% and the predictive value of a positive scan was 46% (Table 3). These results suggest that a ^99m^Tc-HMPAO-labeled leukocyte scan will accurately predict remission near the end of antibiotic treatment. Moreover, based on reports using ^111^In-labeled WBCs, WBC scans appear well suited for monitoring the response of DFO patients to antibiotic therapy over time because the abnormal scan findings revert to normal within two–eight weeks after the start of successful antibiotic therapy [114].

The dual radiopharmaceutical/radiotracer method is a unique strength of planar and SPECT molecular imaging. In the typical application in DFI, one radiotracer labels WBCs while the other targets bone marrow (Figure 8). This approach is particularly powerful in the evaluation of suspected infection superimposed on confounding pathology such as Charcot neuropathic changes. Sixteen to thirty hours after the initial scan, delayed scans of the bone marrow are often performed using a longer-lived tracer if OM is suspected; however, bone deformities such as fracture or abnormal marrow distribution from atypical hematopoietic activity may lead to false positives [114]. To overcome this, visualization of the bone marrow using a second radiotracer (dual-tracer approach) may be adopted. Depending on the tracers used, some dual-tracer approaches allow for simultaneous visualization, while others such as ^99m^Tc-HMPAO require clearance of the first tracer. ^99m^Tc-labeled sulfur colloid is commonly used for this application because the molecule is phagocytized and sequestered by endothelial cells lining the bone marrow, reticular cells, macrophages, and macrophage precursors [115]. A positive diagnosis is indicated by greater or spatially discordant radiotracer uptake on the leukocyte scan compared with the bone marrow scan. Not only does this combined technique improve sensitivity and specificity, but it is also convenient and can be performed within one scan because of the gamma camera’s ability to distinguish the different energies of ^99m^Tc-sulfur colloid and radionuclides such as ^111^In. 

Although the dual-tracer approach is highly effective, it has limitations; principally, localizing the anatomic compartment of both tracers [79]. To improve this technique, WBC-labeled SPECT may be used in conjunction with CT to better visualize anatomically complex areas of the foot. This combined modality approach has been associated with improved distinction between soft tissue infection and OM as well as fewer false positives, resulting in fewer amputations as well as shorter hospital stays compared to anatomical imaging techniques alone [116]. Still, SPECT/CT in combination with labeled-WBC or labeled-WBC and sulfur colloid scan can be expensive, and in vitro leukocyte labelling is limited by the concentration of leukocytes obtained. Similar to planar imaging, radiotracer uptake—and therefore image quality—depends on the number and type of cells labeled as well as the host response to infection.

#### 3.2.3. ^18^F-FDG PET and PET/CT

^18^F-Fluorodeoxyglucose (^18^F-FDG) is a radiolabeled glucose analogue which accumulates in cells that utilize glucose as an energy source. Facilitative glucose transporter proteins bring ^18^F-FDG across the cell membrane to the cytoplasm where it is subsequently phosphorylated by hexokinase. ^18^F-FDG is then trapped in the cell until it is dephosphorylated by glucose-6-phosphatase. The time between retention and clearance reflects the relative activity of glucose transporters, hexokinase, and glucose-6-phosphatase. In comparison to background activity, areas of decreased or absent uptake reflect low or absent glucose metabolism. Regions of increased ^18^F-FDG uptake reflect greater than normal rates of glucose metabolism [117]. Cells with many glucose transporters and a high metabolic rate, such as active inflammatory cells, display high FDG uptake. In addition, the ^18^F-FDG molecule is small and can easily penetrate poorly perfused areas, making it useful for imaging patients with vascular insufficiency [111]. When a dual modality approach of ^18^F-FDG-PET/CT (Figure 10) is adopted, simultaneous CT and PET images are acquired, and a hybrid three-dimensional image is generated that aids in the visualization of infection foci, similar to SPECT/CT imaging. PET-CT is a dual modality that uses the strength of CT and the power of PET tracers to quantify infection and inflammatory processes, and assess many other features of DFI [118]. From its high image resolution and radiotracer sensitivity and specificity, ^18^F-FDG PET/CT is useful in evaluating patients with characteristics that might otherwise compromise the accuracy of MRI or CT, including prior surgery, trauma, or orthopedic hardware [119,120,121]. One concern may be that ^18^F-FDG PET imaging could be compromised in patients with increased blood glucose and concomitant diabetic foot infection. A study by Yang et al. found otherwise. In a clinical evaluation of 21 patients with healthy serum glucose levels (<150 mg/dL) and 27 patients with elevated glucose levels (150–250 mg/dL), ^18^F-FDG PET demonstrated comparable sensitivity for detecting pedal OM between subjects with healthy glucose levels (sensitivity = 87.5%) and subjects with high serum glucose (sensitivity = 88.9%). Impressive levels of specificity of 96.8%, sensitivity of 88.3%, and accuracy of 93.8% were reported for both groups combined [122]. However, DFI patients in a clinical setting often have poorly regulated insulin levels and may have much higher glucose levels. In a retrospective study by Kagna et al., investigators addressed a potential issue that ^18^F-FDG activity may be altered in patients who received antibiotic therapy for a previous infection. The study assessed ^18^F-FDG PET in 176 patients who had received between 1–73 days of antibiotic therapy for an infectious process. They reported no false negative cases, correct identification of infectious foci in 61% of the study population, and true negatives in 29% of the study population; however, the study did not specifically target DFI [123]. 

Although ^18^F-FDG imaging boasts an impressive sensitivity and specificity, it is prone to false positives. Patients with a recent surgical history exhibit increased ^18^F-FDG uptake in the healing tissue because of the increased cellular metabolic rate. Since FDG localizes to sites of increased metabolic activity rather than directly labelling bacteria, ^18^F-FDG results are skewed by the human immune system and cannot differentiate among infection, neoplasia, and sterile inflammation that is seen in Charcot [110].

#### 3.2.4. Gallium Scan

Two isotopes of gallium are in clinical use, namely ^67^Ga and ^68^Ga. Gallium, as an iron mimetic, may be recruited into several processes that localize the isotope to sites of infection. First, gallium creates a complex with the ferric ion delivering glycoprotein transferrin. During uptake of ^67/68^Ga, much of the tracer is bound to circulating plasma transferrin. The increased blood flow and vascular permeability to the site of infection then facilitate the delivery and accumulation of ^67/68^Ga to infectious foci. Any ^67/68^Ga which is not bound to transferrin can also be bound to lactoferrin and circulating leukocytes, be directly taken up by bacteria, or form a complex with bacterial-produced siderophores, which are then transported into the bacterium and phagocytized by macrophages [111,124,125].

Historically, ^67^Ga scans were widely used in institutions with limited access to ex vivo WBC labeling or MRI [79]; however, as the availability of these methods has improved over time, ^67^Ga scans are now mainly used for the imaging of spinal infections. Studies have found other imaging modalities such as Sulesomab-labeled WBC scans (sensitivity = 67%, specificity = 85%) superior to ^67^Ga bone scintigraphy (sensitivity = 44%, specificity = 77%) [70]. ^68^Ga is more commonplace now compared to ^67^Ga. One study assessing the diagnostic potential of ^68^Ga-citrate PET/CT for osteomyelitis reported a sensitivity of 100%, a specificity of 76%, PPV of 85%, NPV 100%, and overall accuracy of 90%; however, this was not categorically evaluated in a DFI model [75]. Another study compared the kinetics of ^18^F-FDG, ^68^Ga-citrate, ^11^C-methionine, and ^11^C-donepezil in a porcine OM model. They found that ^68^Ga-citrate uptake was slow, irreversible, and limited by diffusion whereas the ^18^F-FDG uptake rate is determined by perfusion [126]. 

Because ^68^Ga possesses a shorter half-life and is a positron emitter, it yields higher quality images than those acquired with ^67^Ga. In addition, ^68^Ga may be produced by a generator or cyclotron, while ^67^Ga production requires the use of a cyclotron. As a result, ^68^Ga has displaced ^67^Ga as the gallium isotope of choice in some molecular imaging applications, such as neuroendocrine tumor imaging [127]. Although ^68^Ga retention in infectious sites is, in part, a bacteria-specific localization mechanism (siderophore binding), ^18^F-FDG PET/CT may still have superior sensitivity and specificity due to favorable emission features of ^18^F and glucose utilization by WBCs. However, a recent single institution study found that, for identifying infectious foci in patients with *Staphylococcus aureus* bacteremia, ^68^Ga citrate was comparable to ^18^F-FDG in the detection of osteomyelitis, whereas ^18^F-FDG resulted in a higher signal for detection of soft tissue infection [128]. 

## 4. Emerging Radiotracers in Infection Imaging

With the advancement of existing imaging technologies and the development of new ones, the diagnosis of DFI is becoming increasingly less invasive [129]. Although there have been advances in molecular imaging, further development is needed for targeting the infectious agent(s) rather than the host immune response, as many of the clinically available molecular imaging radiotracers do. Those that do target the microorganism localize to the site of infection by nonspecific mechanisms or are restricted by radiotracer production, availability, or cost. Several developments in the molecular imaging of infections target microorganism-specific metabolism and activity using the imaging modalities discussed in this review [130]. By targeting substances that are presented by or released from the pathogenic microorganism or microorganism-specific metabolic pathways, more radiotracers can be developed whose localization mechanism is independent of the host immune response. Some approaches that are being taken in the development of more specific radiotracers include utilizing radiolabeled antibiotics, antimicrobial peptides, and bacteria-specific agents such as radiolabeled sugars. 

Radiolabeled antibiotics are relatively well-studied in the context of DFI imaging. Previously studied radiolabeled antibiotics target crucial features required for bacterial pathogenicity and survival, including the synthesis of folic acid required for nucleic acid synthesis, cell wall construction, cell membrane structure and function, transcription, and translation [131]. Antibiotic-resistant bacteria pose a major problem for radiolabeled antibiotics. Depending on the uptake mechanism, some bacteria may demonstrate radiotracer accumulation, while others may not. ^99m^Tc-ciprofloxacin is the most studied antibiotic imaging agent and, although previous studies originally demonstrated its high specificity, subsequent studies have not been able to reproduce the result [132,133,134,135]. In vivo studies of its ability to differentiate gram-positive from gram-negative bacteria have reached contradictory conclusions. The main drawback to ^99m^Tc-ciprofloxacin is its accumulation in dead bacteria [132], which leads to false positives, as suggested by a large multicenter clinical trial that also reported a sensitivity of 66.7%, a specificity of 85.7%, and an accuracy of 72% for both soft tissue infection and OM [136]. Several radiolabeled antibiotics have been studied to date (Table 4); however, the least well-studied are bacterial cell wall synthesis inhibitors such as vancomycin. The SPECT tracer ^99^ Tc-vancomycin, which was directly labeled, has demonstrated an affinity for *S. aureus* infectious foci. ^99m^Tc-vancomycin labeled through ^99m^Tc-Tc-HYNIC tetrazine click chemistry corroborated this observation with a three-fold uptake increase in a *S. aureus* infection site compared to controls [137]. Studies using fluorescently labeled vancomycin have also reported promising results [138], although isotopic-labeled vancomycin has shown greater uptake [139]. Although the ^99m^Tc-vancomycin SPECT tracer may have a valuable place in the clinical management of DFI, the development of a vancomycin PET tracer may provide better image resolution. 

Perhaps the most promising approach is to target the metabolism of the microorganism itself. One approach targets bacteria-specific transport systems for sugars. For example, sorbitol is used by gram-negative bacteria and has been developed as a tracer and been successfully studied in humans. One study, using ^18^F-fluorodeoxysorbitol (FDS) in humans, reported selective uptake in infected lesions, low background uptake, and fast plasma clearance [140]. However, uptake by *Enterobacteriaceae* of the gut was also seen. A more recent study of ^18^F-FDS demonstrated the sorbitol-specific pathway by observing uptake in clinically relevant strains of *Enterobacterales.* Following the in vitro study and a previous in vivo study in murine models, they acquired ^18^F-FDS PET/CT images in humans, demonstrating the safety and ability of ^18^F-FDS to differentiate infection from sterile inflammation [141]. In addition, it was used to monitor antibiotic treatment [140]. Bacterial cell wall amino sugars, present in gram positive as well as gram negative bacteria, have also been targeted for imaging. One study in rats used 2-deoxy-2-[^18^F]fluoroacetamido-D-glucopyranose (^18^F-FAG) to successfully identify *E. coli* infection from sterile inflammation by imaging with confirmation by histology [142]. Perhaps the most effective of the previously discussed radiolabeled sugars are maltodextrin-based imaging probes. The maltodextrin transport pathway is bacteria-specific and can discriminate between live bacteria, metabolically inactive bacteria, and inflammation induced by lipopolysaccharides. PET imaging performed with ^18^F-maltohexose demonstrated uptake in both gram positive and gram negative bacteria [143], although poor signal-to-noise ratios limit this tracer’s clinical application. The second generation tracer 6′′-^18^F-fluoromaltotriose has improved signal-to-noise ratios while maintaining bacterial-selective uptake in vitro and in vivo [144,145]. 

Amino acid uptake by bacteria has also been targeted for radiotracer development. D-stereoisomers of amino acids are found in bacterial cell walls but are not present in humans, making these amino acid isoforms promising pathogen-specific imaging targets. One analog of D-methionine, D-[methyl-^11^C]methionine, has been used to differentiate sterile inflammation from *E. coli* and *S. aureus* infections in a murine model, reporting an SUV_max_ 8–10 times higher (SUV_max_ = 0.8% ID/cc) in *E.coli* and *S. aureus* compared to [^11^C]L-methionine (SUV_max_ = 0.1% ID/cc) [146]. An in vitro study demonstrated the broad sensitivity of D-[methyl ^11^C] methionine across a panel of clinically relevant DFI pathogens, including *E. coli*, *P. aeruginosa*, *P. mirabilis*, *S. aureus*, *S. epidermidis*, and *E. faecalis,* with the greatest uptake occurring in *S. aureus* and *P. aeruginosa* [147]. A recent study compared the uptake of D-5-[^11^C]glutamine and L-5-[^11^C]glutamine isomers in a murine dual pathogen myositis model with localized MRSA and *E.coli* infections. Uptake of L-5-[^11^C]glutamine was higher in the heat-killed control site when compared to D-5-[^11^C]glutamine, indicating that the D-stereoisomer had better pathogen specificity. When D-5-[^11^C]glutamine was compared to ^18^F-FDG in a sterile inflammation model, ^18^F-FDG uptake was significantly higher, further demonstrating the specificity of D-5-[^11^C]glutamine for bacterial metabolism [148]. Although ^11^C-labeled amino acid-based imaging shows specificity, widespread adoption for clinical use may be limited because of the 20-min half-life of ^11^C. 

Microorganisms may also be targeted based on their vitamin uptake. Since biotin is used in the production of fatty acids and is an essential growth factor for *S. aureus* and other bacteria, it has been explored for infection imaging. Although little data are currently available, an in vitro study using [^111^In]-In-DTPA-biotin reported selective uptake in *S. aureus* cultures [149]. In addition to targeting biotin, a radiolabeled vitamin B12 derivative, ^99m^Tc-PAMA has also been used for infection imaging. Since PAMA accumulates in rapidly proliferating cells, the imaging agent has shown high uptake in *S. aureus* as well as *E. coli* in vitro and in vivo [150]. Microorganisms commonly use a compound called PABA to synthesize folic acid. A study using ^3^H-PABA reported accumulation in MRSA as well as other therapy resistant microorganisms. Uptake was 100× greater in the microorganisms than in mammalian cells. When labeled with ^18^F, uptake was seen in *S. aureus* [151]. 

Other targets for bacterial imaging include nucleoside analogs, peptidoglycan targeting aptamers, siderophores, and radiolabeled antibodies. The nucleoside analog ^124^I-fialuridine (^124^I-FIAU) is a bacterial tyrosine kinase substrate trapped in the cell following phosphorylation. Although it has been previously used to detect prosthetic joint infections [152], a more recent study found that tracer uptake was reduced in the presence of metal artifacts [153]. A clinical study of 16 patients with suspected musculoskeletal infection, who underwent ^18^F-FDG and ^124^I-FIAU PET/CT, reported a range of specificity of 51.7–72.7% and a maximum sensitivity of 100%; however, the minimum sensitivity was incalculable. The authors were also unable to come to a definitive conclusion on its diagnostic efficacy because of equivocal clinical findings in a large number of patients [154]. Aptamers are oligonucleotides that have been used to target the peptidoglycan cell wall. One group that used ^99m^Tc-Antibac1 demonstrated its ability to distinguish between bacterial and fungal infections in vivo and in vitro [155].

Siderophores are small molecules with a high affinity for Fe(III) and are produced primarily by bacteria, but are also produced by virtually all microorganisms—including prokaryotes and eukaryotes [156]. Mammalian macrophages have been reported as sources of siderophores [157]. Organisms use these metal-seeking molecules for extra- or intra-cellular metal transport and storage. Biological processes such as DNA replication, transcription, oxidative stress responses, and respiration all require transition metals. Iron and zinc are among the most abundant in living organisms. Iron is an essential cofactor for respiration and central metabolism. Zinc is the catalytic center for many enzymes and is required for 5–6% of all protein functions [158]. Siderophores are often categorized into the following groups according to the chemical moiety that binds iron: catecholates, hydroxamates, and carboxylates. Mixed type siderophores have two or more of these moieties. Siderophores may actively alter their affinity for iron in order to bind other metals by increasing their expression of metal transporters [158]. Desferricoprogen—a natural siderophore produced by *Penicillium chrysogenum* and *Neurospora crass*—reportedly bound to both trivalent (Al(III) and In(III)) or divalent (Cu(II), Ni(II), Zn(II), and Fe(II) metal ions [159]. Transport systems and siderophore production are unregulated during infection and can accumulate in bacteria as well as fungi. An in vitro study which used ^68^Ga-ferrioxamine (^68^Ga-FOXE) showed high uptake in *S. aureus*; however, it was not corroborated in vivo [160]. In fungi, *Aspergillus fumigatus*-specific fluorescent siderophore conjugates were used to assess pulmonary aspergillosis in vitro [161]; however, recent in vivo studies have only been performed to assess biodistribution in pulmonary aspergillosis [162]. Further research regarding specificity, sensitivity, and use in additional infection models is needed.

In addition to in vitro labeling, WBCs may also be localized in vivo by using radiolabeled antibodies which target WBC antigens—a technique called immunoscintigraphy. For example, ^99m^Tc-besilesomab uses the monoclonal antibody besilesomab to bind to the granulocyte-specific antigen-95. Antigen-95 is present on the membranes of granulocytes and their precursors. Within 45 min of initial injection, 10% of the ^99m^Tc-besilesomab is bound to neutrophils, while 20% is unbound but localizes to the site of infection by nonspecific mechanisms [163]. A recent study of 119 patients with suspected OM compared planar ^99m^Tc-HMPAO WBC and ^99m^Tc-besilesomab imaging in the diagnosis of peripheral OM. They found that ^99m^Tc-HMPAO had a lower sensitivity than ^99m^Tc-besilesomab (59.0% and 74.8%, respectively); however, ^99m^Tc-HMPAO had a higher specificity than ^99m^Tc-besilesomab (79.5% and 71.8%, respectively) [63]. ^99m^Tc-sulesomab also binds to an antigen found on leukocyte membranes—antigen-90—but only 3–6% of ^99m^Tc-sulesomab is bound to circulating neutrophils, and 35% is found in bone marrow 24 h post-injection [111]. The results of a study of combat-related infections suggest that ^99m^Tc-sulesomab may not possess sufficient accuracy (78% accuracy) for the identification of the foci of infections, but may have otherwise comparable diagnostic values (93% PPV, 62% NPV, 72% sensitivity, 88% specificity) [71]. One study of *Yersinia enterocolitica* infections used immunoPET with ^64^Cu-NODAGA-labeled Yersinia-specific polyclonal antibodies to target the membrane protein YadA and reported colocalization in a dose-dependent manner with bacterial lesions in a murine model [164]. Another study that used the [^64^Cu]NODAGA-IgG_3_ monoclonal antibody from mice to examine *C. albicans* infections demonstrated its accuracy in diagnosing infection in vitro as well as in vivo [165]. A meta-analysis found a sensitivity of 81% and specificity of 86% for anti-granulocyte scintigraphy with monoclonal antibodies [166]. The use of intact antibodies takes up to several days to accumulate near infectious foci. Clinically, this could be a limitation because of the need for rapid identification of the infection site. However, the use of antibody fragments may lead to faster accumulation time. Overall, it appears that radiolabeled antibodies do not have the ability to replace existing methods in a clinical setting because of the relatively low diagnostic values reported by the studies discussed above. 

One recent therapeutic application of US used in DFI is cavitation, in which ultrasonic waves are used to create microbubbles which form from the dissolved gas that accumulates in the wound. The molecular purpose of this treatment is to induce compression and movement of the wound cells and small ions to increase protein synthesis and the permeability of vascular walls and cell membranes [133]. Ultimately, reduced inflammation, angiogenesis, and increased cell proliferation and recruitment are expected at the site of infection [134]. Currently, only three studies have used this technique and two of them demonstrated either complete wound closure or a significant reduction in wound area [135,136,137].

Although these emerging imaging probes are targeting the microorganism rather than the host response, the quantitative measures of probe retention are generally much lower than the clinically available tracers that target the host response. Further, while most of these have not been used in DFI specifically, this is an opportunity for the expansion of molecular imaging approaches to DFI.

## 5. Conclusions

Currently, the clinical diagnosis of diabetic foot soft tissue infection or OM relies on corroboration by multiple clinical tests, including the probe-to-bone test, pathology, and microbiology. Many of these techniques require special training or are prone to low inter-rater reliability [170]. While imaging modalities such as plain film radiography and MR have demonstrated reasonable agreement (62%) [170], the area of infection is only identified after morphological alteration. In general, molecular imaging approaches possess higher sensitivity and specificity and can indicate pathological changes well before morphological changes occur. Although radiotracers, such as ^18^F-FDG or those that are used in cellular carriers such as WBCs, are useful in assessing diabetic foot infection, the development of radiotracers that are microorganism-specific is of strong interest. This may have utility in assessing complex cases or treatment efficacy. A novel focus of interest is directly targeting osteoblasts and osteoclasts, both of which can be infected by microorganisms. As a result, changes have been observed in the production of proteins for immune modulation, bone resorption, and intra-cellular signaling. Undoubtedly, the infected cells must have an altered cell surface protein milieu that can be targeted for a new kind of molecular imaging assay.

## Figures and Tables

**Figure 1 ijms-22-11552-f001:**
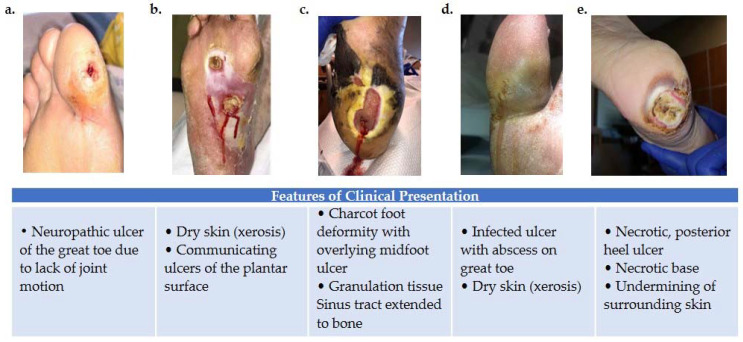
Variations in foot wound location, presentation, and severity as illustrated by images (**a**–**e**). (**a**) Patient presenting with neuropathic ulcer under the proximal interphalangeal joint of the hallux. The ulcer is related to lack of joint motion at the first metatarsophalangeal joint. (**b**) Patient presenting with an infected ulcer following the flexor tendons of the foot. Notice a blow lesion at the plantar arch. Dry skin (xerosis) is a sign of autonomic neuropathy. (**c**) Patient presenting with Charcot foot deformity and overlying midfoot ulcer. Macerated skin around the edges of the ulcer and a sinus tract that extends to the bone is also seen. (**d**) Patient with an infected ulcer with abscess on the great toe and xerosis suggesting autonomic neuropathy. (**e**) Patient with a posterior heel ulcer containing a necrotic base and undermining of surrounding skin.

**Figure 2 ijms-22-11552-f002:**
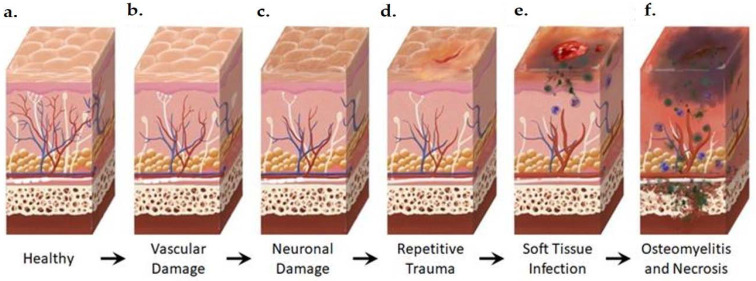
Anatomical Changes Associated with Diabetic Foot Pathophysiology. Diabetic neuropathy is a multi-faceted polyneuropathy related to an increased risk of ulceration, infection, and amputation. Sustained hyperglycemia damages the endothelial lining of the blood vessels in previously healthy tissue (**a**), leading to impaired circulation (**b**). Without sufficient vascular support, nerves die off and the skin may become dry and cracked as sweat secretions decrease (**c**). In the event of injury, numbness in the foot due to neuronal ischemia may mean that insults go undetected for some time (**d**). Fissures in the dried skin can harbor microorganisms, increasing the likelihood of wound infection. Initial microbial invasion of the trauma site leads to inflammation, vasodilation, and soft tissue necrosis (**e**). Decreased vascularization compromises immune response to infection and prolongs healing time. If the infection persists, usually because of delayed care or ineffective treatment, microbes may invade bone tissue, leading to osteomyelitis and bone deformation (**f**). White: neurons, red: arteries, blue: veins, purple: polymorphonuclear lymphocytes, green: microorganisms.

**Figure 3 ijms-22-11552-f003:**
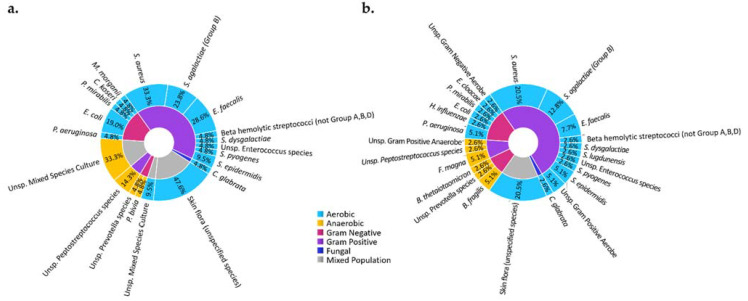
Microbiota of DFI of observed in subjects of an ongoing clinical study in a tertiary care facility in our medical center at the University of Texas Southwestern Medical Center. Microbiota of DFI in soft tissue biopsies (**a**) and bone biopsies (**b**) show that most microbes are aerobic (blue) and gram positive (purple), with Staphylococcus aureus being the most identified microorganism. The study was conducted according to the guidelines of the Declaration of Helsinki and approved by the Institutional Review Board (or Ethics Committee) of the University of Texas Southwestern Medical Center (IRB: 112016-043, approved November 2016). Informed consent was obtained from all subjects involved in the study.

**Figure 4 ijms-22-11552-f004:**
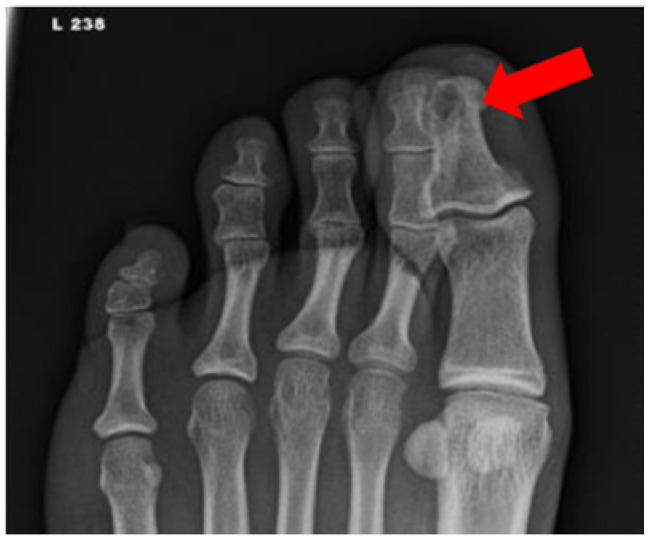
Radiograph of a foot in a diabetic patient with a history of trauma to the great toe. Anteroposterior view of the left foot demonstrates soft tissue swelling and focal osteolysis to the distal phalanx of the great toe (arrow) with periostitis.

**Figure 5 ijms-22-11552-f005:**
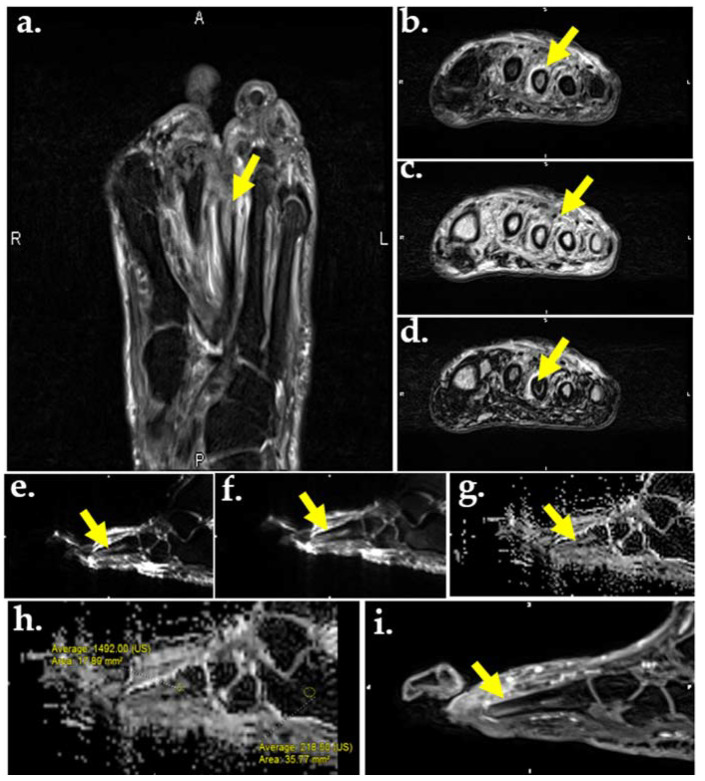
MR images of the foot in a 56-year-old male with DM and plantar ulcer with plantar ulcer below the 3rd metatarsal and suspected OM. Prior fracture deformity of the second metatarsal head and surgical resection of the third metatarsal are noted. T2-weighted fat suppressed image (**a**) and T2 Dixon water map (**b**) show marrow edema of the third metatarsal stump (arrows). In-phase T2 Dixon map (**c**) shows muscle fatty replacement from DM denervation change (arrow). Opposed-phase T2 Dixon map (**d**) shows marrow involvement by OM. DWI images (**e**,**f**) show marrow replacement by OM on low (**a**), high (**b**), and apparent diffusion coefficient (ADC) maps (**g**,**h**). Quantification of the marrow abnormality by ADC (**h**) measures 1.49 compared to normal marrow of 0.21, indicating no intra-osseous abscess, which would appear dark on ADC and bright on DWI. (**i**) Contrast-enhanced MR shows enhancement of the metatarsal stump.

**Figure 6 ijms-22-11552-f006:**
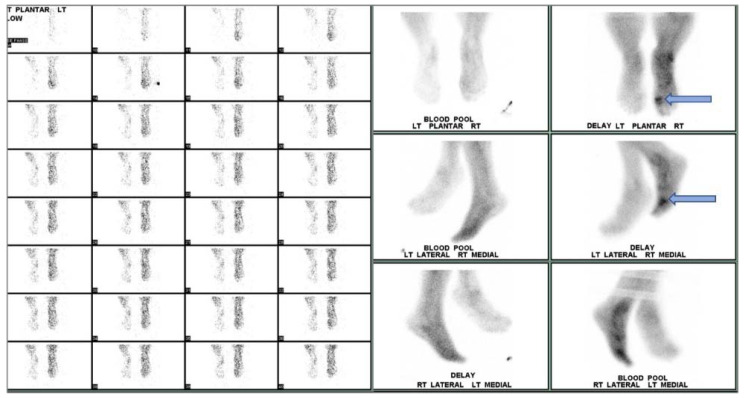
Three-phase bone scan of an individual with confirmed DFI. Cellulitis is visualized in the right foot with OM in the first metatarsophalangeal region (**arrow**).

**Figure 7 ijms-22-11552-f007:**
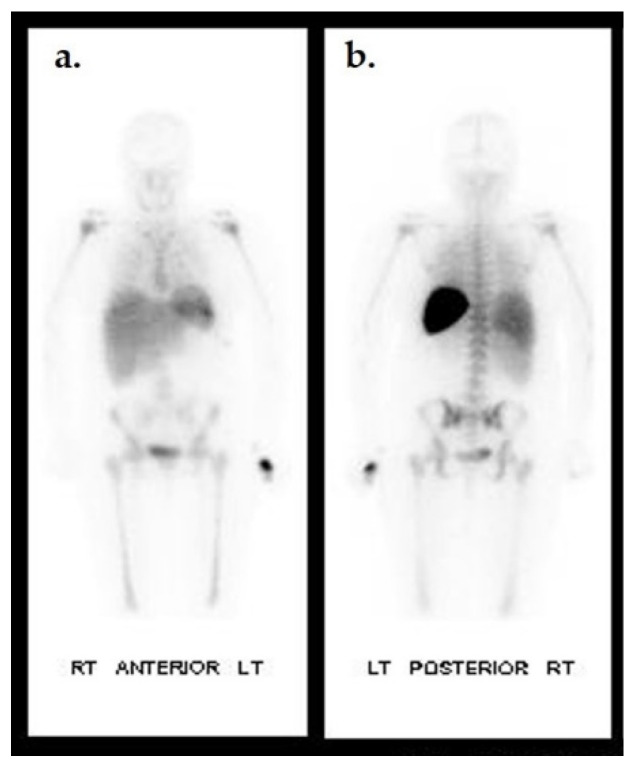
Biodistribution of ^111^In-oxine WBC in a non-infected individual. Anterior (**a**) and posterior (**b**) planar images of a healthy patient 24 h after ^111^In-oxine-labeled WBC injection. Uptake is seen in the spleen, liver, and bone marrow.

**Figure 8 ijms-22-11552-f008:**
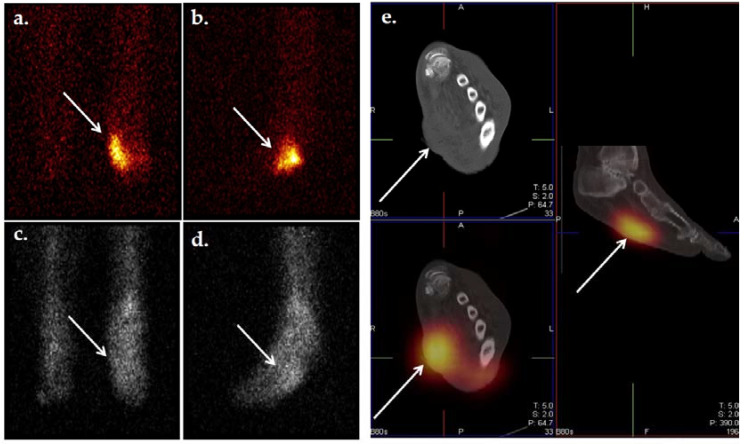
Dual tracer imaging using ^111^ n-WBC and ^99m^Tc-sulfur colloid. Planar ^111^In-WBC (**a**,**b**) and ^99m^Tc-sulfur colloid (**c**,**d**) images from a 62-year-old male with diabetes with a left foot abscess. There is spatial and intensity discordance in activity from the radionuclides. Anterior (**a**) and lateral (**b**) ^111^In-WBC images show focus of increased activity in the left mid foot. Anterior (**c**) and lateral (**d**) ^99m^Tc-sulfur colloid images show diffuse activity throughout the mid and hind foot, suggesting the development of Charcot foot. Axial and sagittal ^111^In-WBC SPECT/CT (**e**) localized activity to an abscess in the plantar aspect of the left mid foot. Osteomyelitis was excluded.

**Figure 9 ijms-22-11552-f009:**
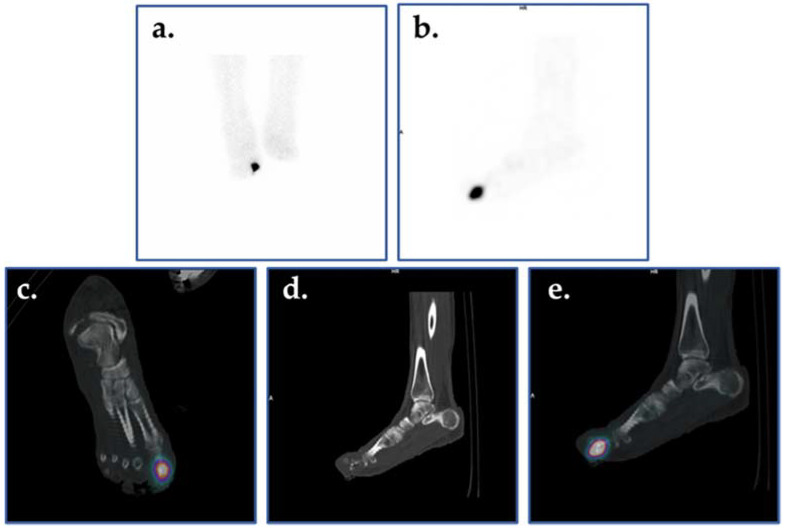
Planar scintigraphic and SPECT/CT ^99m^Tc-WBC images in an individual with DFI. The patient’s presentation was suspicious for OM involving the great toe. Planar images (**a**,**b**) demonstrate increased radiolabeled WBCs in the right forefoot, perhaps in the region of the toes. SPECT/CT (**c**–**e**) allows for precise localization of infection to the proximal phalanx of the right great toe.

**Figure 10 ijms-22-11552-f010:**
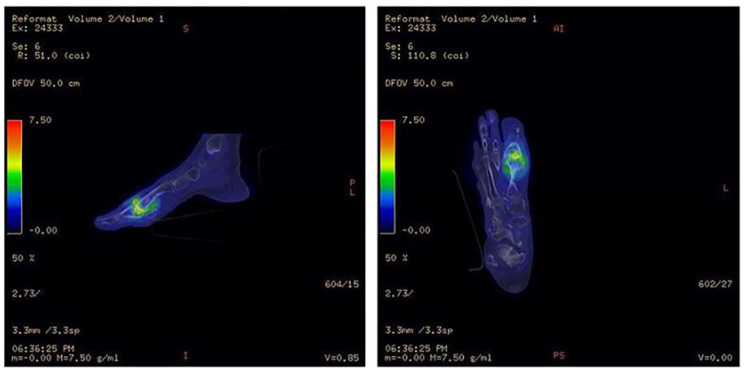
^18^F-FDG PET/CT illustrating infection of the metatarsophalangeal joint. Increased ^18^F-FDG uptake in the first metatarsophalangeal joint is seen in the bone and soft tissues. Signal uptake in this region is much greater than that of the surrounding healthy tissue, indicating pathologically increased glucose metabolism. Image has been reproduced with permission from Iyengar et al., J Clin Orthop Trauma, published by Elsevier, 2021 [117].

**Table 1 ijms-22-11552-t001:** Diagnostic Imaging Modalities for Diabetic Foot Infections.

Imaging Method/Radionuclide		Sensitivity (%)	Specificity (%)	Function	Cost	Accessibility	Radiation	References
Radiography		43–86	27–83	+++	+	+++++	☢	[50,51,52,53,54]
CT		67–80	50–70	+++	+++	+++	☢	[50,55]
Ultrasound		79	80	++	+	+++++	O	[56]
MRI		87–100	37–83.8	++++	+++	+++	O	[52,53,54,57,58,59,60,61,62]
Planar Bone Scintigraphy	^99m^Tc-MDP	81.0–84.2	28.0–67.7	+++	++	++	☢☢☢	[52,54]
Planar WBC Scan	^99m^Tc-besilesomab	74.8	71.8	+++	+++	++	☢☢☢	[63]
	^99m^Tc-HMPAO	59.0–91	79.5–92	+++	++	++	☢☢☢	[63,64]
	^99m^Tc-exametazime	86.0	100	+++	++	++	☢☢☢	[65]
WBC SPECT/CT	^99m^Tc-WBC	87.5–100	35–92	++++	++++	+	☢☢☢☢	[58,62,66,67,68,69]
	^99m^Tc-sulesomab	67–72.0	85–88.0	+++	+++	++	☢☢☢☢	[70,71]
	^111^In-WBC	74–92	68–75	++++	++++	+	☢☢☢☢	[54,58]
PET	^18^F-FDG	81–92.3	92.0–93	+++	++++	+	☢☢☢	[72,73]
PET/CT	^18^F-FDG	43–89	67–100	++++	++++	+	☢☢☢☢	[58,65]
	^67^Ga-citrate	44–100	45–77	++	++	++	☢☢☢	[70,74]
	^68^Ga-citrate	100	76	++	++	++	☢☢☢	[75]

Note: Relative radiation level ratings for effective adult dose: O = 0 mSv, ☢ < 0.1 mSv, ☢☢☢ = 1–10 mSv, ☢☢☢☢ = 10–30 mSv. Utility of each technique was adapted from Noguerol et al. and is graded from most useful (+++++) to least useful (+) [76]. Radiation ratings are adapted from the 2019 ACR appropriateness criteria [49].

**Table 2 ijms-22-11552-t002:** Summary of Molecular MR Imaging Techniques in the Assessment of the Diabetic Foot.

MR Technique	Molecular Basis	Demonstrated Imaging Feature of DFI
Dynamic Contrast Enhancement (DCE)	Contrast agent alters MR signal intensity in a concentration dependent manner	Bone marrow edema Pattern of Soft tissue involvement Joint impairment Vascular involvement
Diffusion Weighted Imaging (DWI)	Takes advantage of restricted diffusion in certain anatomical features such as abscesses and compares this to free water to provide an enhanced image with excellent background suppression.	Bone marrow edema Soft tissue involvement Joint impairment Nerve damage
Dixon Sequence	Combines in-phase and out-of-phase images produced through chemical-shift with decreased sensitivity to inhomogeneities of B0 and B1, resulting in homogenous fat suppression. Cortical margins and cysts are best seen on out-of-phase image, marrow edema on water-image, muscle fatty replacement and marrow fat replacement on in-phase and fat-images.	Bone marrow edema Bone lesion identification Nerve damage
Diffusion Tensor Imaging (DTI)	Uses the sensitivity of DWI to the anisotropic water movement within myelinated axons to generate high resolution images that can provide information regarding myelin sheath and axonal damage	Nerve damage

**Table 3 ijms-22-11552-t003:** Common Molecular Imaging Radiotracers.

Radiotracer	Imaging Technique	Cellular Parameter	Mechanism of Localization	Strengths	Weaknesses
^99m^Tc-MDP	Bone Scintigraphy	Osteoblastic bone formation	Chemiabsorption onto hydroxyapatite crystals of the bone matrix	Inexpensive High sensitivity	Low specificity, dependent on area of exposed bone surface
^18^F-NaF	Bone Scintigraphy or PET	Osteoblastic bone formation	Binds to and engages exchange reaction with hydroxyapatite crystals to form hydroxyfluoroapatite and fluoroapatite in the bone matrix	Smaller molecule than MDP with faster uptake, fast renal clearance, less background	Dependent on area of exposed bone surface
^99m^Tc-HMPAO	Labeled WBC (in vitro)	WBC migration to site of infection	WBC response to infection	Specificity, same day diagnosis of DFI	Simultaneous dual-tracer approach is not possible. Intensive preparation Expensive
^111^I-oxine	Labeled WBC (in vitro)	WBC migration to site of infection	WBC response to infection	Simultaneous dual-tracer approach is possible	Intensive preparation Expensive Low image quality
^18^F-FDG	PET/CT	Glucose uptake and metabolism	Taken up by WBC or immune cells with increased glucose disposal	Sensitivity, inherently tomographic	Specificity, Availability, and cost
^67/68^Ga-Citrate	Scintigraphy/SPECT/CT (^67^Ga) or PET/CT (^68^Ga)	WBC activity at site of infection	Iron mimetic. Binds to transferrin in circulating plasma. Binds to lactoferrin released from dying WBCs and bacterial siderophores at the site of infection	Detects low grade infection Low toxicity, Bone/soft tissue distinction	Delayed imaging High radiation dose, Non-specific accumulation in sterile inflammation or osteoblasts in healing bone

**Table 4 ijms-22-11552-t004:** Emerging Investigational Radiotracers of Infection Imaging.

	Radiotracer	Clinical Trials	Parameter	Mechanism of Localization	Strength/Weakness	References
**Radiolabeled Antibiotics**	^99m^Tc-ciprofloxacin	Yes	Inhibition of DNA Synthesis	Bacterial DNA gyrase	High sensitivity (85.4–97.2%), ciprofloxacin already used in DFI treatment	[133,134,135,136]
					Low specificity (66.7–81.7%), antibiotic resistant bacteria	
	^18^F-fluoropropyl-trimethoprim	No	Inhibition of Folic Acid Synthesis	Inhibition of thymidine biosynthesis	Low background, high uptake in bacteria, detect inflammation from soft tissue infection vs sterile inflammation	[167]
					Antibiotic resistant bacteria	
	^99m^Tc-sulfonamides (pertechnetate, sulfadiazine)	No	Inhibition of Folic Acid Synthesis		Broad spectrum antibiotics, uptake in bacterial and fungal infections	[155]
					Antibiotic resistant bacteria	
	^99m^Tc-vancomycin	No	Inhibition of bacterial cell wall synthesis	Binds to D-ala-D-ala lipid moiety	Specific for gram positive organisms	[137,138,139]
					Not specific for gram negative organisms Antibiotic resistant bacteria	
**Radiolabeled Sugars**	^18^F-FDS	Yes	Bacteria-Specific Glucose Sources for Carbohydrate Metabolism	Bacterial Metabolic Substrate	Antibiotic treatment monitoring, used in humans	[140,141]
					Uptake by Enterobacteriaceae in the human gut	
	^18^F-FAG	No		Sorbitol analogue utilized only by bacteria	Selective accumulation in E. coli, rapid accumulation, can differentiate infection from sterile inflammation, shows promise for monitoring response to treatment, small molecule	[142]
					Not applied clinically	
	^18^F-maltohexose	No		Bacterial-specific maltodextrin transporter	Can discriminate between live bacteria, metabolically inactive bacteria, and sterile inflammation	[143]
					Poor signal-to-noise ratios, Not applied clinically	
	6′′-^18^F-fluoromaltotriose	No		Bacterial-specific maltodextrin transporter	2nd Gen, improved signal-to-noise ratio, bacterial-selective uptake in vitro and in vivo	[144,145]
					Not applied clinically	
**Amino Acid Uptake**	D-[methyl-^11^C] methionine	No	Bacterial Cell Wall Synthesis	Incorporation into the peptidoglycan	Distinguish sterile inflammation from infection in both gram—and gram +, broad sensitivity	[146,147]
					Not applied clinically	
	D-5-[^11^C] glutamine	No		Incorporation into the peptidoglycan	Highly specific, high sensitivity for gram +, no uptake in sterile inflammation, fast clearance	[148]
					Corroborating studies needed, not yet applied clinically	
**Vitamin Uptake**	^124^I-fialuridine (FIAU)	Yes	Endogenous TK enzyme of pathogenic bacteria	Trapped in the cell after phosphorylation	Reduced uptake in the presence of metal artifacts,	[154,168,169]
					More clinical studies needed to assess clinical efficacy	
	^111^In-biotin	No	Production of Fatty Acid	Bacterial growth factor	Essential growth factor for S. aureus	[149]
					Corroborating in vivo studies needed to assess clinical relevance	
	^99m^Tc-PAMA	No	Vitamin B12 Metabolism	Vitamin B12 derivative that accumulates in rapidly proliferating cells	High uptake in Gram + and Gram -	[150]
					Not applied clinically	
	^18^F or ^3^H-PABA	No	Folic Acid Synthesis	Inhibition of Thymidine Synthesis	Accumulation in MRSA and other resistant organisms	[151]
					In vivo studies needed	
**Polyclonal Antibodies**	^64^Cu-NODAGA	No	Membrane protein binding of polyclonal antibody	Microbe-specific membrane polyclonal antibody binding	Particular to a specific microbe	[164,165]
					Slow accumulation time	
**Siderophores**	^68^Ga-FOXE	No	Iron Transport	Accumulation of Siderophores in the cell	High uptake in S. aureus and fungi	[160,161,162]
					Not used in DFI model	
**Immunoscintigraphy**	^99m^Tc-sulesomab	Yes	WBC migration to infectious foci	Binds to antigen-90 on WBC membranes	Ease of preparation? Not sure about this	[63,71]
					Dependent upon host response, expensive, limited availability	
	^99m^Tc-Besilesomab			Binds to antigen-95 on granulocytes and their precursors	Ease of preparation, good sensitivity and specificity	[63]
					Dependent upon host response, expensive, limited availability	

Table adapted from Ankrah et al. (2018) [131].

## Data Availability

Not applicable.

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
