# Peer review of "Pathophysiology and Molecular Imaging of Diabetic Foot Infections"

_ijms, 2021, doi:10.3390/ijms222111552_

Round 1

Author Response

The response has been submitted

Reviewer 2 Report

# General comments:
1. I hope the authors can label the source of the figures.
2. The authors spent a lot of space to introduce the Molecular Imaging. Such as the CT, MRI, ultrasound, and angiography. But the summary of molecular images is not mentioned.
# Specific comments:
1 .Add the reference of "Historically, classification foot infections." in line 135
2. Add the reference of "Poorly controlled …osmotic stress." in line 170
3. Add the reference of "Such information can …detection threshold." in line 242.
4. For the state-of-the-art, the authors may consider the refer the elastography analysis as below article: Using Elastographic Ultrasound to Assess Plantar Tissue Stiffness after Walking at Different Speeds and Durations, Applied Sciences 10, no. 21: 7498

Author Response

The point-by-point response is contained in the submitted MS Word document.

Round 2

Reviewer 1 Report

All the remarks I made in the first edition of the manuscript have been made by the authors.
I was initially concerned that a method that is overall, albeit statistically, the most effective had not been proposed. This could be included in the conclusions of the manuscript in some way or even remain as a future challenge to the next researcher or a future conclusion from the specific research team, which is very familiar with this scientific field.